# Circadian ABCG2 Expression Influences the Brain Uptake of Donepezil across the Blood–Cerebrospinal Fluid Barrier

**DOI:** 10.3390/ijms25095014

**Published:** 2024-05-03

**Authors:** André Furtado, Ana Catarina Duarte, Ana R. Costa, Isabel Gonçalves, Cecília R. A. Santos, Eugenia Gallardo, Telma Quintela

**Affiliations:** 1CICS-UBI, Health Sciences Research Centre, Faculty of Health Sciences, Universityof Beira Interior, Avenida Infante Dom Henrique, 6200-506 Covilhã, Portugal; 2Laboratório de Fármaco-Toxicologia-UBIMedical, Universidade da Beira Interior, 6200-506 Covilhã, Portugal; 3Faculty of Health Sciences, Instituto Politécnico da Guarda, 6300-559 Guarda, Portugal

**Keywords:** circadian rhythm, blood–cerebrospinal fluid barrier, ABCG2, Donepezil

## Abstract

Donepezil (DNPZ) is a cholinesterase inhibitor used for the management of Alzheimer’s disease (AD) and is dependent on membrane transporters such as ABCG2 to actively cross brain barriers and reach its target site of action in the brain. Located in the brain ventricles, the choroid plexus (CP) forms an interface between the cerebrospinal fluid (CSF) and the bloodstream, known as the blood–CSF barrier (BCSFB). Historically, the BCSFB has received little attention as a potential pathway for drug delivery to the central nervous system (CNS). Nonetheless, this barrier is presently viewed as a dynamic transport interface that limits the traffic of molecules into and out of the CNS through the presence of membrane transporters, with parallel activity with the BBB. The localization and expression of drug transporters in brain barriers represent a huge obstacle for drug delivery to the brain and a major challenge for the development of therapeutic approaches to CNS disorders. The widespread interest in understanding how circadian clocks modulate many processes that define drug delivery in order to predict the variability in drug safety and efficacy is the next bridge to improve effective treatment. In this context, this study aims at characterizing the circadian expression of ABCG2 and DNPZ circadian transport profile using an in vitro model of the BCSFB. We found that ABCG2 displays a circadian pattern and DNPZ is transported in a circadian way across this barrier. This study will strongly impact on the capacity to modulate the BCSFB in order to control the penetration of DNPZ into the brain and improve therapeutic strategies for the treatment of AD according to the time of the day.

## 1. Introduction

Alzheimer’s disease (AD) is the most common cause of dementia in humans. It is a primary degenerative disease and is clinically characterized by memory loss, the impairment of multiple cognitive functions and dementia. Among the few drugs approved by the US Food and Drug Administration for the treatment of AD, Donepezil (DNPZ), a second-generation acetylcholinesterase inhibitor, is widely used for the treatment of mild, moderate and severe AD [1]. Currently, DNPZ is used as a standard symptomatic treatment, with an important contribution to slowing disease progression. However, the side effects resulting from the higher doses prescribed to contradict the insufficient drug delivery to the brain are limiting the efficacy of medication for AD treatment [2]. In fact, brain barriers are restricting the drug access to the brain contributing to the low bioavailability [3].

The central nervous system (CNS) maintains the homeostasis of the brain by the presence of two main barriers: the blood–brain barrier (BBB), which separates the systemic circulation from the CNS; and the blood–cerebrospinal fluid barrier (BCSFB) that separates the cerebrospinal compartment from the blood circulation [4]. The BCSFB is established by tight junctions in the epithelium of the choroid plexus (CP), and the arachnoid membrane. It is in contact with the cerebrospinal fluid (CSF) in the apical side and there is a vast network of fenestrated capillary blood vessels on its basal side [5]. Since it may regulate the distribution of certain drugs and neurotoxic agents between the blood and the CSF, the CP epithelium of the BCSFB is considered a pharmacologic and toxicologically important barrier [6]. The delivery of drugs to the CNS is primarily impeded by the BBB. However, we are not alone in our view that the BCSFB could be distinguished as a potential gateway to the brain due to its arquitecture, strategic position and highly dynamic transport activity [7]. The CP has been associated with numerous functions, including CSF production and secretion, chemical surveillance and detoxification of the CSF and barrier function, as it constitutes a selective physical barrier to the passage of compounds, toxins, cells and molecules to and from the CNS [8,9,10,11,12,13,14].

As a selective barrier between the bloodstream and the CSF, CP epithelial cells display several membrane transporter proteins, tight junctions and detoxification enzymes, enabling CP cells to control the traffic of molecules across the BCSFB [15]. ATP-binding cassette (ABC), solute carrier (SLC) and solute carrier organic anion (SLCO) transporter families are expressed in the CP [12]. Depending on the transporters, they can be responsible for the efflux or uptake of molecules by CP cells [12,16,17], affecting the pharmacokinetics of multiple therapeutic drugs [18]. Therefore, drug transporters are in part responsible for the drug delivery into the brain and the effective drug concentrations at the target tissue. At the BCSFB, multiple transporters have been identified, namely ABCG2 (BCRP), which is expressed on the apical/subapical side of the BCSFB, contributing to the transport of its substrates to the CSF [5,10,19]. ABCG2 was described as a transporter for DNPZ in heart and brain tissues, with a possible clinical application [20]. Curiously, the location of ABCG2 at the apical BCSFB side, will imply a function that will be the opposite of what occurs at the blood–brain barrier, involved in limiting the drug distribution to the CNS [21].

The circadian clock has been demonstrated to interact with drug transporters, influencing the pharmacokinetics and pharmacodynamics of the drugs, reducing side effects and improving therapeutic potential [22]. Recently, it was demonstrated that CP epithelial cells that compose the BCSFB harbor a functional circadian clock, considered an important component of the circadian clock hierarchy [11,23]. In addition, the circadian expression of BCSFB transporters, such as ABCC1, ABCG2, ABCC4 and OAT3, was described in the CP of Wistar rats [10]. Hence, the widespread concern in understanding how circadian rhythms modulate many processes of drug transport into the brain in order to predict the variability in drug safety and efficacy is the next step to overcome brain barriers and improve effective treatments [24]. Thus, characterizing the circadian regulation of ABCG2 at the BCSFB and the circadian profile of DNPZ transport across the barrier will strongly impact on the capacity to modulate the BCSFB in order to control the penetration of DNPZ into the brain.

## 2. Results

### 2.1. Circadian Expression of rBMAL1 and rABCG2 in CPEC

The circadian expression of the clock gene *rBMAL1* was assessed to confirm the synchronization of CPEC primary cultures. The results show a significant circadian variation (CircWave_V1.4, www.hutlab.nl, *p* < 0.05), with peak expression at around 7 h after synchronization (Figure 1).

The daily profile of *rABCG2* membrane transporter was also evaluated and showed a significant circadian variation (CircWave, *p* < 0.05), with a peak at expression around 19 h after synchronization (Figure 2).

### 2.2. Circadian Profile of DNPZ Transport in an In Vitro Model of the BCSFB

The presence of the circadian expression of ABCG2 in synchronized CPECs led us to examine if the transport of DNPZ across an in vitro model of the BCSFB is circadian-dependent. According to the results shown in Figure 3, the DNPZ concentration in the apical compartment oscillated (CircWave, *p* < 0.05), with a peak at approximately 22 h after synchronization (Figure 3). In the basal and cell compartment, no significant oscillation in the DNPZ concentration was observed (Figure 3).

### 2.3. The Role of ABCG2 in the Transport of DNPZ across the BCSFB

To study the role of ABCG2 in the circadian transport of DNPZ across an in vitro model of BCSFB, a transport assay was carried out using an ABCG2 inhibitor (Ko143). We found that in the presence of Ko143, the DNPZ concentration in the apical compartment lost the circadian rhythmicity observed in its absence (Figure 4). On the contrary, in the basal compartment, the DNPZ concentration oscillated (CircWave, *p* < 0.05), with a peak at around 12 h after synchronization (Figure 4). Within the intracellular compartment, circadian variation was also observed in the DNPZ concentration (CircWave, *p* < 0.05), with a peak between 8 and 9 h after synchronization (Figure 4).

## 3. Discussion

The BCSFB is found in the CP of the ventricular system of the brain, and works as a highly selective barrier to the passage of molecules from the bloodstream to the CSF, providing an obstacle to the delivery of therapeutics for CNS disorders. Consequently, this security system is responsible for the deficient brain bioavailability of several pharmacological agents [25]. Thus, the development of delivery strategies to cross brain barriers requires a better knowledge of the circumstances that change the drug concentration–time profile in the brain. Among the several factors that have been studied, the circadian rhythm, particularly the modulation of membrane transporters according to the time of day, is increasingly shaping our view of future research in the improvement of AD therapy. In approaching this issue, in the present study, we analyze the effects of the circadian clock in ABCG2 expression and its possible involvement in the transport of DNPZ across the BCSFB. Our results confirmed, for the first time, that the circadian expression of ABCG2 controls DNPZ bioavailability in the CSF.

In our study, we observed that ABCG2 displays a circadian pattern in CPEC primary cultures. This result was not surprising, as in our previous work, we also showed that ABCG2 displays a circadian pattern of expression on the female rat CP and in synchronized a human choroid plexus papilloma cell line (HIBCPP) [10]. Additionally, in other tissues such as the mouse small intestine and liver, ABCG2 also displayed circadian rhythmicity [26,27]. In the small intestine, ABCG2 rhythmic expression was described as being dependent of the molecular clock as Clock-mutant mice lacked ABCG2 rhythmic expression [26]. In the liver, ABCG2 circadian expression was lost in Per1 and Per2 double-transgenic mice [27].

DNPZ is an approved therapy for the treatment of AD. Its transport across brain barriers is dependent of membrane transporters which play an important role in drug disposition and toxicity [28]. ABCG2, SLC5A7, SLC22A1, SLC22A2, SLC22A3, SLC22A4 and SLC22A5 have been identified as membrane transporters for DNPZ [20,28]. Furthermore, ABCG2, SLC22A2, SLC22A3, SLC22A4 and SLC22A5 have all been described at the BCSFB [29,30]. ABCG2 is located in the apical/ subapical choroidal membrane in human and mouse CP cells and contributes to the transport of DNPZ into the CSF [5,31]. SLC22A2 and SLC22A3 are located in the apical membrane of the CP and mediate the entry of molecules into the cell cytoplasm [32]. SLC22A4 and SLC22A5 localization is still imprecise, with reports showing their expression in the apical, as well as basal membrane of the CP, and they are responsible for the choroidal uptake of their respective substrates [32,33,34].

To characterize the role of ABCG2 in the circadian transmembrane transport of DNPZ, an in vitro model of the BCSFB using primary cultures of CPECs was implemented. Interestingly, we observed daily oscillations in DNPZ concentrations in the apical compartment. Peak ABCG2 expression at the CPEC occurs around 19 h, 2 h before the peak expression of DNPZ transport across the BCSFB. This finding points to the idea that an increase in the ABCG2 expression at a specific time point of the day will intensify the activity of ABCG2, and DNPZ is transported at an increased rate into the CSF. In the basal and intracellular compartments, the circadian transport profile of DNPZ was not observed. The role of ABCG2 as a multidrug transporter that affects drug pharmacokinetics was also demonstrated in the intestine. ABCG2 circadian oscillation in mouse intestine was a determinant for the circadian bioavailability of an ABCG2 substrate, sulfasalazine [26]. More recently, the involvement of murine ABCG2 in the transport and secretion of melatonin metabolites to the intestine and kidney also highlights the idea that changes in ABCG2 expression might affect melatonin therapeutic activity [35]. Targeting ABCG2 regulation to successfully deliver drugs to the brain was also widely considered at the blood–brain barrier over the past few years. In fact, the increase in transporter expression and/or activity at the blood–brain barrier was studied to treat neurological disorders, as well as protect the brain [16].

The specificity of ABCG2-mediated circadian transport of DNPZ across the BCSFB was further studied, carrying out the inhibition of the efflux transporter with Ko143. Interestingly, the abolishment of the daily oscillations of DNPZ transport in the apical compartment and the emergence of oscillations in the basal and intracellular compartments was observed. This finding may point to an alternative mechanism of DNPZ transport across the BCSFB. As mentioned before, DNPZ is transported by other BCSFB membrane transporters. Of these, two were described as possibly located in the basal compartment of CP, i.e., SLC22A4 and SLC22A5.Furthermore, both genes were considered the circadian targets for several drugs, namely SCL22A4 for Androgel and Combivent, and SLC22A5 for Lidoderm, Niaspan and Combivent [36]. Considering these observations, we hypothesized that SLC22A4 and SLC22A5 might contribute to the circadian oscillation of DNPZ concentrations in both the basal and intracellular compartments. Among these two transporters, SLC22A4 was involved in the suppression of oxidative stress in PC12 cells through the uptake of an antioxidant under saturable conditions [37]. In addition, the circadian expression of mouse SLC22A4 induces dosing–time-dependent differences in the absorption of gabapentin from the intestine [38].

Thus, it will be possible that with the inhibition of ABCG2, an increase in DNPZ bioavailability in basal and intracellular compartments might be sufficient to trigger a saturable transport of DNPZ by SLC22A4, with circadian oscillations taking place in the basal and intracellular compartments.

Despite this interpretation, the involvement of ABCG2 in the transport of DNPZ into the CSF is crucial to optimize therapy. This assumption is corroborated by a recent study that measured DNPZ concentrations in the plasma and CSF of AD patients at four different time points and calculated the plasma/CSF ratio. They observed that the plasma and CSF DNPZ levels increase over 24 h without statistical significance. Interestingly, the plasma/CSF ratio significantly increased overtime [37], pointing to the importance of CSF DNPZ concentration levels to optimize DNPZ dosage.

In summary, the present findings reveal the regulation of ABCG2 expression by the circadian rhythm which impacts the circadian-dependent transport of DNPZ across the BCSFB into the CSF. It will be important to mention that in situations where several drugs are administered, it is possible that competition for the DNPZ receptor may occur, which might affect the rhythmicity of the drug’s transport. The present study will have important implications in the modulation of the BCSFB in order to control the penetration of DNPZ into the CSF. Consequently, it will provide powerful data to optimize AD therapy, taking into account the circadian clock, increasing efficacy and reducing side effects commonly associated with some pharmacological interventions. Predicting a priori the most appropriate timing when DNPZ transport equilibrium occurs at the BCSFB to match the therapeutic approach to the patient characteristics will be important for the treatment of AD, improving personalized medical interventions.

## 4. Material and Methods

### 4.1. Animals

This study was conducted with the approval of the Animal Welfare and Ethics Committee of the Health Science Research Centre of the University of Beira Interior, in compliance with the National and European Union rules for the care and handling of laboratory animals. The CPs were collected from the lateral ventricles of 2–7-day-old postnatal *Wistar Han* rats, which were housed in appropriate cages at constant room temperature in a 12 h light/12 h dark photoperiod and given standard laboratory chow and water ad libitum. Efforts were made to minimize the number of animals and animal suffering.

### 4.2. Choroid Plexus Epithelial Primary Culture

Thirty postnatal animals were anesthetized on ice for at least 30 min before being euthanized. The CPs were collected from the lateral ventricles and used to establish choroid plexus epithelial cell (CPEC) primary cultures as previously described by Gonçalves et al. [39]. Briefly, the dissociated cells were seeded into 6-well culture plates and cultured in a high-glucose DMEM supplemented with5 µg/mL insulin (Sigma-Aldrich, Merck, Algés, Portugal), 100 U/mL penicillin, 100 µg/mL streptomycin, 10% *v*/*v* fetal bovine serum (FBS), 10 ng/mL epidermal growth factor (Sigma-Aldrich, Merck, Portugal) and 30 µM cytosine arabinoside (Sigma-Aldrich, Merck, Portugal). The cultures were maintained in a humidified incubator at 37 °C and 5% CO_2_. The culture medium was replaced at day in vitro 1 (DIV1) and every 2 days thereafter. All studies were conducted using cultures established for 4–5 days.

#### 4.2.1. ABCG2 Circadian Pattern

CPEC primary cultures established for at least 4–5 days were trypsinized and seeded in 24-well culture plates (approximately 1.5 ×10^4^ cells/well). The culture medium was changed every 2 days and experiments were conducted 8 days after seeding. CPECs were synchronized with 100 nM dexamethasone (Sigma-Aldrich, Merck, Portugal), an artificial glucocorticoid that resets the circadian clock in the culture, for 2 h. The culture medium was changed, and cells were collected 4 h after synchronization and then every 4 h during a 24 h period for total RNA extraction.

#### 4.2.2. Quantitative Real-Time PCR (qPCR)

Total RNA was isolated from the CPEC using triple Xtractor reagent (Grisp, Porto, Portugal) according to the manufacturer’s instructions. Total RNA purity and integrity were assessed by the measurement of the absorbances at 260 and 280 nm using a NanoPhotometer^TM^ (Implen, Munich, Germany). NZY M-MuLV Reverse Transcriptase (NZYTech Ltd.,Lisboa, Portugal), Random hexamer mix (NZYTech Ltd., Portugal), GRS dNTP mix (GRISP Ltd., Portugal) and RNA (500 ng) were used for cDNA synthesis following manufacturer’s instructions.

qPCR was performed to assess the daily expression of rBmal and *rABCG2*. Rat cyclophilin A (rCyc) was used as a housekeeping gene. Primers sequences are listed in Table 3. qPCR was performed using a CFX-ConnectTM Real-Time PCR Detection System (Bio-Rad, Hercules, CA, USA) using an Xpert Fast SYBR 2x mastermix (Grisp, Porto, Portugal). The qPCR protocol consisted of an initial 3-min denaturation step at 95 °C, followed by 40 cycles of 5 s at 95 °C, 30 s at 62 °C and 10 s at 72 °C. The transcripts amplification was validated by the profiles of melting curves. All primers were previously tested with the following cDNA dilutions: stock, 1:2, 1:4, 1:8. To calculate ΔCt we used the average of the Cts of a sample from which we subtracted the average of the Cts of the housekeeping gene for the same sample. To calculate ΔΔCt we first subtract the average of the ΔCt from the average of the Cts of the housekeeping gene of all the samples. Finally, we subtract the previous result from the ΔCt of the desired sample.

### 4.3. Donepezil Transport Assay

The DNPZ transport assay in CPEC was determined to investigate whether the daily oscillations in *rABCG2* would affect the transport of its substrate across the BCSFB. CPEC primary cultures established for at least 4–5 days were trypsinized and seeded in a transwell filter system and used as an in vitro model of the BCSFB. Cell culture inserts apical compartment were previously coated with collagen following Monnot A.Det al. protocol [42]. CPEC were seeded on the apical compartment of cell culture inserts (pore diameter 0.4 µm and insert area 0.33 cm^2^; VWR, Alfragide, Portugal) at a density of 2.5 × 10^4^ cells per insert in high-glucose DMEM supplemented with 5 µg/mL insulin, 100 U/mL penicillin, 100 µg/mL streptomycin, 10% *v*/*v* FBS, 10 ng/mL epidermal growth factor and 30 µM cytosine arabinoside. The culture medium was changed every 2 days. The paracellular permeability was assessed every day by the measurement of TEER values using an Epithelial Volt/Ohm Meter (WPI, Sarasota, FL, USA). On the 8th day of culture, TEER values reached 65–80 Ω·cm^2^. Two other criteria of membrane integrity were used: appearance of a confluent monolayer on the insert under the microscope; height of the culture medium on the inner chamber (had to be at least 2 mm higher than that on the outer chamber for at least 24 h) [40]. At this point, cells were synchronized with 100 nM dexamethasone for 2 h. The culture medium was changed, and the cells were placed in a humidified incubator at 37 °C and 5% CO_2_. At 4 different time points after synchronization (4, 10, 16, 22 h) cells were washed 3 times with Krebs-Ringer buffer (KRB) and placed in the incubator in KRB for another hour. Next, cells were incubated on the apical side with DNPZ 40 μg/mL (Sigma-Aldrich, Merck, Portugal) for 3 h. After incubation, both the apical and basal mediums were collected and the remaining cells were washed 3 times with KRB, trypsinized and also collected.

The role of *rABCG2* in the transport of DNPZ across the BCSFB was analyzed, using a similar assay with an inhibitor of *rABCG2* (Ko143 100 nM; Tebu-bio, Lisbon, Portugal).

#### 4.3.1. DNPZ Quantification

The quantification of DNPZ was performed on an HPLC 1290 with a binary pump coupled to a fluorescence multi-wavelength detector (FLD-3400 RS detector), both from Agilent Technologies (Soquímica, Lisboa, Portugal). The separation was performed in an anEclipse Plus C_18_ (3.5 μm, 4.6 × 100 mm) analytical column from Agilent Technologies (Soquímica, Lisboa, Portugal) protected by a pre column. The flow rate was 1 mL/min, with a mobile phase consisting of 0.1% methanol as solvent A and acetic acid (70:30, *v*/*v*) as solvent B. The elution was carried out in gradient mode. The temperature of the sampler was set at 4 °C and the injection volume was 20 µL. The determination of DNPZ was performed at λex 269 nm and λem 390 nm (Figure 5).

#### 4.3.2. Validation Procedure

The described method was validated according to the guiding principles of the Food and Drug Administration [43], within the linearity range between 0.04 and 40 μg/mL, and an additional four quality controls (0.04, 1.25, 10, and 40 μg/mL) (n = 3) included. The criteria used to assess the fitness of this linear model included a weighted determination coefficient (R^2^) higher than 0.99, and the accuracy of the calibrators within ±15% from the nominal value (except at the LLOQ, where ±20% was accepted) was adopted as the acceptance criteria. The method’s LLOQ was defined as the lowest concentration that could be precisely and accurately measured, with a coefficient of variation (CV) equal to or lower than 20% and a relative error (RE) within ±20% of the nominal concentration. To evaluate sensitivity, the limit of detection (LOD) as a signal-to-noise ratio >3 was calculated, with 10 replicates performed at a concentration of 0.04 μg/mL.

Precision and accuracy were evaluated during the 5-day protocol, adopting the same concentrations used for the quality controls. Coefficients of variation (CV) equal to or lower than 15% were accepted for precision at all studied concentration levels, while for accuracy, a mean relative error (RE) of ±15% (from the nominal concentration) was accepted for all concentrations, except the LLOQ (±20%).

The CVs obtained in the study of inter-day precision and accuracy (RE) were typically lower than 8%, with an accuracy ranging from 0.3 to 3.4% (Table 4). As for intra-day precision and accuracy, it was evaluated on the same day by the analysis of five replicates at 0.04 (LLOQ), 1.25, 10 and 40 μg/mL. The obtained CVs were once again within the accepted criteria, with the CVs lower than ±14%, and accuracy ranging from 0.5 to 14% (Table 4). The outliers were calculated taking into account these FDA criteria at the level of each parameter, considering the permitted CV and RE.

Parameters that were evaluated consisted of selectivity, linearity, limit of quantification (LLOQ), limit of detection (LOD), accuracy and precision. The selectivity method was evaluated in the presence of potential interferences (e.g., compounds of the medium of culture) and no signals at the retention time of DNPZ were detected. Linearity was tested in the concentration range from 0.04 (LLOQ) to 40 µg/mL. The acceptance criteria of acceptance included the determination coefficient value (R^2^) > 0.99, as well as the calibrators’ accuracy within a ±15% (except at the LLOQ, where ±20% was considered acceptable). Table 5 shows the calibration data.

### 4.4. Statistical Analysis

A normality test was performed to ensure a normal distribution (Shapiro–Wilk normality test). CircWave v1.4 analysis software (Dr. Roelof A. Hut) was used to analyze the rhythmicity of *rBMAL1*, *rABCG2* and DNPZ concentrations in all three compartments (apical, basal and cells) by a harmonic regression method, with an assumed period of 24 h and an alpha set at 0.05 for the expression data, and an assumed period of 48 h and an alpha set at 0.05 for the transport data. Oscillations were considered statistically significant when *p* < 0.05. The outliers were calculated using the ESD method (extreme Studentized deviate) with an alpha = 0.05.

## Figures and Tables

**Figure 1 ijms-25-05014-f001:**
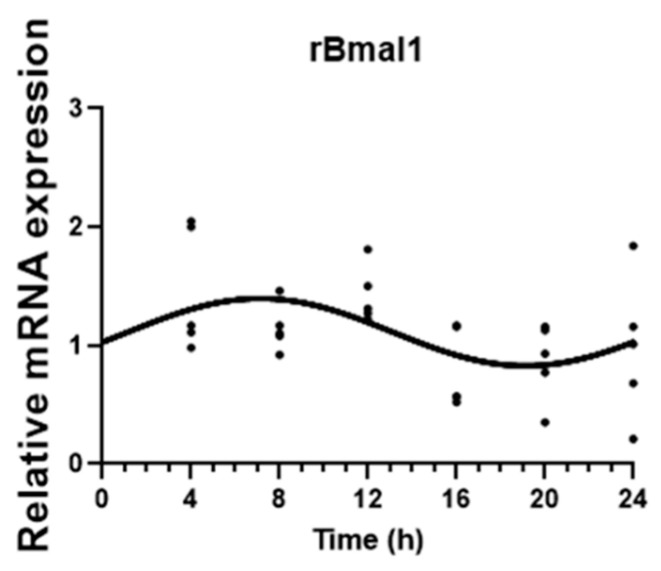
CircWave analysis of rat basic helix–loop–helix ARNT-like 1 (*rBMAL1*) clock gene circadian expression in CPEC. *rBMAL1* expression was analyzed every 4 h during a 24 h period in synchronized CPECs. The sine–cosine fit represents a significant 24 h period oscillation (*p* < 0.05). Statistical analysis is shown in Table 1.

**Figure 2 ijms-25-05014-f002:**
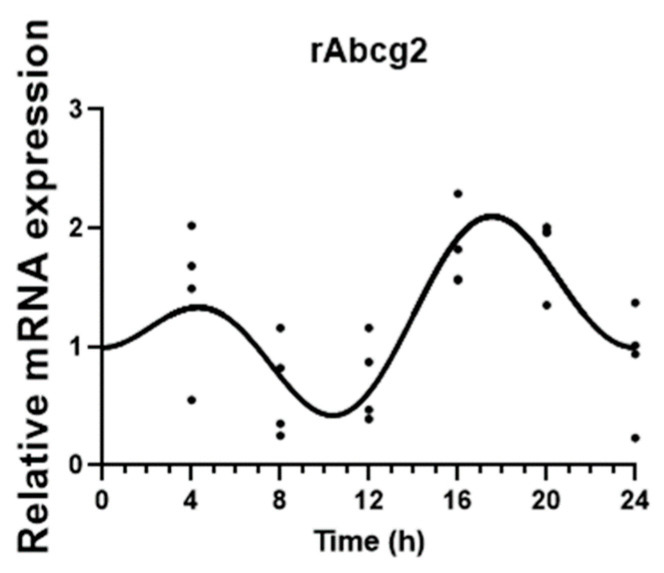
CircWave analysis of rat *rABCG2* membrane transporter gene circadian expression in CPEC. *rABCG2* expression was analyzed every 4 h during a 24 h period in synchronized CPECs. The sine–cosine fit represents a significant 24 h period oscillation (*p* < 0.05). Statistical analysis is shown in Table 1.

**Figure 3 ijms-25-05014-f003:**
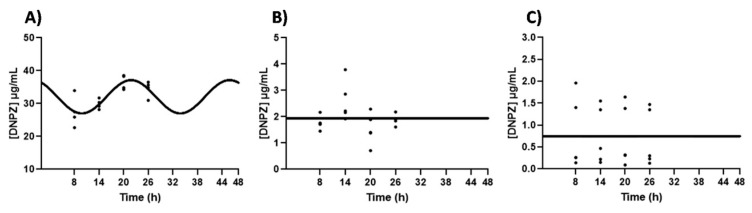
DNPZ transport across an in vitro model of the BCSFB. CircWave analysis of DNPZ levels in apical (**A**), basal (**B**) and intracellular (**C**) compartments. The represented curves indicate a statistically significant rhythm (CircWave, *p* < 0.05). Statistical analysis is shown in Table 2.

**Figure 4 ijms-25-05014-f004:**
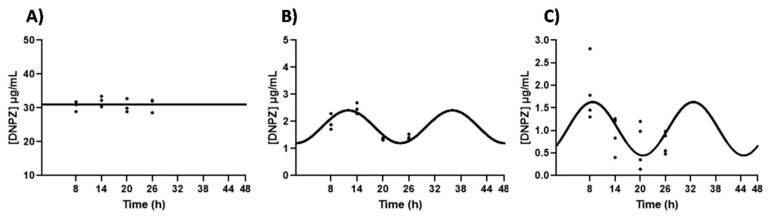
Effects of ABCG2 inhibition in the DNPZ circadian transport across an in vitro model of the BCSFB. CircWave analysis of the DNPZ levels from apical (**A**), basal (**B**) and intracellular (**C**) compartments after the inhibition of ABCG2. The represented curves indicate a statistically significant rhythm (CircWave, *p* < 0.05). Statistical analysis is shown in Table 2.

**Figure 5 ijms-25-05014-f005:**
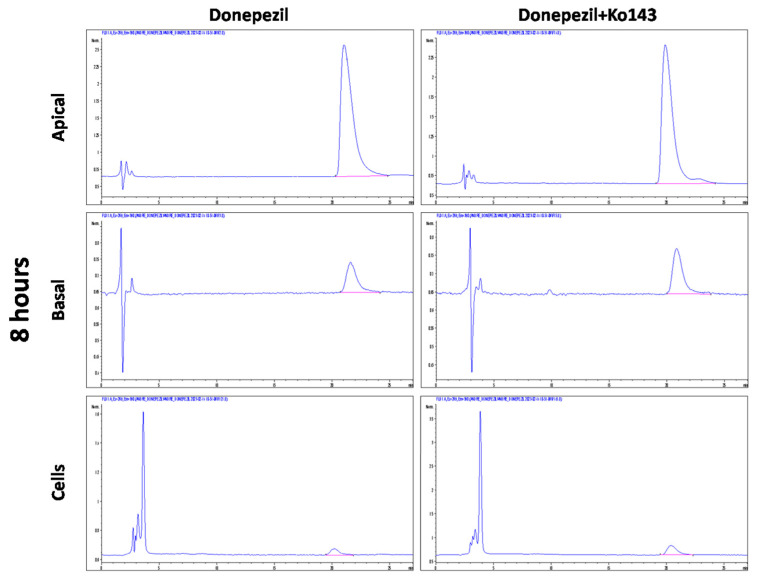
Representative chromatogram obtained for Donepezil (10 µg/mL) and Donepezil + inhibitor (Ko143) in apical, basal and cell compartments. Chromatogram for Donepezil (10 µg/mL) and Donepezil + inhibitor (Ko143) at λex 269 nm and λem 390 nm.

**Table 1 ijms-25-05014-t001:** Significance (*p*-value) and center of gravity (COG) values for *rBMAL1* and *rABCG2* determined by CircWave analysis.

Gene	*rBMAL1*	*p*-value = 0.0418 COG = 7.13
	*rABCG2*	*p*-value = 0.0012 COG = 19.52

**Table 2 ijms-25-05014-t002:** Significance (*p*-value) and center of gravity (COG) values for DNPZ concentrations in an in vitro model of the BCSFB determined by CircWave analysis.

DNPZ	Apical	*p*-value = 0.0010 COG = 21.82
	Basal	*p*-value > 0.05 COG = 12.78
	Cells	*p*-value > 0.05 COG = 10.86
DNPZ + Ko143	Apical	*p*-value > 0.05 COG = 13.85
	Basal	*p*-value = 0.007 COG = 12.01
	Cells	*p*-value = 0.0224 COG = 8.66

**Table 3 ijms-25-05014-t003:** Primers used in RT-PCR and Real-time qPCR.

Gene	Primers	Bp	Ref.
*rBmal1*	FW-ACACTGCACCTCGGGAGCGA RV-CGCCGAGCTCCAGAGCACAA	100	[40]
*rABCG2*	FW-GGCCTGGACAAAGTAGCAGA RV-CACAGTTGTGGGCTCATCCAGGAA	141	[41]
*rCyc*	FW-CAAGACTGAGTGGCTGGATGG RV-GCCCGCAAGTCAAAGAAATTAGAG	163	[9]

**Table 4 ijms-25-05014-t004:** Inter-day (n = 5), intra-day (n = 5) precision and accuracy.

Concentration (μg/mL)	Inter-Day Precision			Intra-Day Precision		
	Measured *	CV (%)	RE	Measured *	CV (%)	RE
0.04	0.04 ± 0.003	7.24	3.94	0.04 ± 0.005	13.78	−2.00
1.25	1.22 ± 0.050	4.07	−2.40	1.24 ± 0.056	4.53	−0.48
10	9.97 ± 0.652	6.54	−0.32	11.35 ± 0.616	5.43	13.54
40	39.80 ± 2.195	5.52	−0.51	40.96 ± 2.256	5.51	2.40

* Mean values ± standard deviation.

**Table 5 ijms-25-05014-t005:** Linearity data (n = 5), LOD and LLOQ (n = 10).

Linear Range (μg/mL)	Linearity		R^2 a^	LOD (μg/mL)	LLOQ (μg/mL)
	Slope ^a^	Intercept ^a^			
0.04–40	4.088 ± 0.034	0.050 ± 0.131	0.9994 ± 0.0004	0.04	0.04

^a^ Mean values ± standard deviation.

## Data Availability

Data is available on request.

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
