# Peer review of "Circadian ABCG2 Expression Influences the Brain Uptake of Donepezil across the Blood–Cerebrospinal Fluid Barrier"

_ijms, 2024, doi:10.3390/ijms25095014_

Round 1

Reviewer 1 Report

Comments and Suggestions for Authors

Authors  study aims at characterize the circadian expression of Abcg2 and  profile od Donepezil circadian trasport using an in vitro model of the 23 BCSFB. and improve therapeutic strategies for 26 the treatment of AD. Final purpose is to improve therapy for the treatment of Alzheimer's disease.  

Introduction and references seem complete. Materials ahd methods and discussion should be inverted. Methods seem correct. Conclusions are unremarkable. Authors could also consider if a possible competition of Donepezil with other drugs competing for receptor could disrupt circadian pattern.

Author Response

The authors have carefully revised the paper following the suggestions of the reviewers. We hope we were able to meet all the referees’ suggestions and clarify their doubts, and that the paper is now suitable to meet the International Journal of Molecular Sciences standards. Answers to the specific comments/suggestions/queries are the following:

Reviewer 1

Authors study aims at characterize the circadian expression of Abcg2 and profile of Donepezil circadian transport using an in vitro model of the BCSFB and improve therapeutic strategies for the treatment of AD. Final purpose is to improve therapy for the treatment of Alzheimer's disease. 

Introduction and references seem complete. Materials and methods and discussion should be inverted. Methods seem correct. Conclusions are unremarkable. Authors could also consider if a possible competition of Donepezil with other drugs competing for receptor could disrupt circadian pattern.

AR: We thank the reviewer for the suggestions. The organization of the manuscript is according to the journal’s guidelines for manuscript preparation. Regarding the competition of Donepezil with other drugs, in our experiment this hypothesis is not in question, since our experiment was conducted only with Donepezil and with all the necessary negative controls. However, in situations where several drugs are administered, it is possible that competition for the Donepezil receptor may occur, which might affect the rhythmicity of the drug's transport. In future, it would be interesting to study the effect of simultaneous drugs in the circadian transport of Donepezil. This hypothesis was included in the discussion section.

Reviewer 2 Report

Comments and Suggestions for Authors

The Circadian Expression of Abcg2 Influences the Brain Uptake 2 of Donepezil Across the Blood Cerebrospinal Fluid Barrier  André Furtado et al

Dear Authors

The study evaluates the DNPZ transport assay in CPEC in rAbcg2 and how affect the transport of its substrate across the BCSFB. However, this method does not really reflect the blood brain barrier penetrance since astrocyte, endotelial cells and other cell type influence penetrance of drugs in the brain, which does not really occur in this model.

My Decision is reject

This model does not represent a circadiam regulation of these markers because  the evaluation by PCR are not appropited in this design. The model also does not stimate the real aim of this study

-The in vitro model has many limitations and it can not be considered a in vitro model of

-PCR are not valid for this aim. Protein activities could reflect a real effect are not appropiated take into account the design. We should determinate protein levels, which represent the active protein with functional effects. This PCR are not representative of the aim of the study

-Figures for HPLC should show all treatements, including the inhibitor.

-How these authors evaluate the rhythmicity of rBmal1, rAbcg2 and the DNPZ concentrations in all three compartments  (apical, basal and cells) by a harmonic regression method withevaluated the CircWave v1.4 analysis software. Please, give details of this method.

The statistical analiysis por and it is not clear. The graph indicate 4 points by hour. However, ANOVA, Kruskal Wallis or not parametric analysis are not present.

-The graph does not have statistical errors or media error standard. Please, correct it.

Line 257. Please, explain (details) about how The ΔCt was calculated using the housekeeping gene as the reference gene, and the ΔΔCt was calculated between the normalized ΔCt values from each time point and the average Ct of all the time points tested.

Line 257. Please, explain the reason by which we use 100nM dexamethasone for the celular synchronizedion.

The role of rAbcg2 in the transport of DNPZ across the BCSFB was analyzed, using  a similar assay with an inhibitor of rAbcg2 (Ko143 100nM; Tebu-bio, Lisbon, Portugal). Please, indicate all graphs of HPLC for treatments.

-Indicate details on the procedure for the validation of the described procedure by the principles of the Food 302 and Drug Administration, cyte 26 [26].
Explain how outliders were calculated in your study

Minnor comments

Line 299. The cromatogram for DNPZ . Chromatogram obtained for Donepezil (10µg/mL). Chromatogram for Donepezil  (10µg/mL) must also indicate curves with the inhibitor (show representative figures)

Comments on the Quality of English Language

Extensive englist revision is required

Author Response

The authors have carefully revised the paper following the suggestions of the reviewers. We hope we were able to meet all the referees’ suggestions and clarify their doubts, and that the paper is now suitable to meet the International Journal of Molecular Sciences standards. Answers to the specific comments/suggestions/queries are the following:

Reviewer 2

Dear Authors

The study evaluates the DNPZ transport assay in CPEC in rAbcg2 and how affect the transport of its substrate across the BCSFB. However, this method does not really reflect the blood brain barrier penetrance since astrocyte, endotelial cells and other cell type influence penetrance of drugs in the brain, which does not really occur in this model.

My Decision is reject

This model does not represent a circadian regulation of these markers because  the evaluation by PCR are not appropited in this design. The model also does not stimate the real aim of this study

-The in vitro model has many limitations and it can not be considered a in vitro model of

AR: The choroid plexus is the major component of the BCSFB. The in vitro BCSFB model used in our experiments is widely described and consists of a 2D barrier model where the choroid plexus epithelial cells are cultured on a cell culture inserts. The choroid plexus epithelial cells form a tight confluent monolayer, which is considered an active cellular barrier. The epithelium is in direct continuity with the ependymal layer lining the ventricle, and unlike BBB, the BCSFB located at the epithelial level, is much more permeable that BBB. Drug distribution into the CSF is a function of transport across the CP and the BCSFB is now considered a crucial barrier to access the brain [1]. Considering these unique characteristics, we choose the validated 2D barrier model to study the relative expression of transporters at the CP and the drug transport across the CP epithelium to the CSF.

-PCR are not valid for this aim. Protein activities could reflect a real effect are not appropiated take into account the design. We should determinate protein levels, which represent the active protein with functional effects. This PCR are not representative of the aim of the study

AR: We agree with the reviewer that the PCR analysis should be complemented with protein expression. However, in our study, the rhythmic transporters expression at the RNA level were complemented with in vitro functional studies with the transport of the target drug, which in our view is much more important than determine protein levels. Several studies have already drawn functional conclusions with only RNA analysis, which have then been complemented by studies or computer models to evaluate circadian rhythm function [2-4].

-Figures for HPLC should show all treatements, including the inhibitor.

AR: We have included representative figures in the manuscript.

-How these authors evaluate the rhythmicity of rBmal1, rAbcg2 and the DNPZ concentrations in all three compartments  (apical, basal and cells) by a harmonic regression method withevaluated the CircWave v1.4 analysis software. Please, give details of this method.

AR: Circwave software fits one or more fundamental sinusoidal curves through the individual data points and compares this with a horizontal line through the data mean (a constant). If the fitted curve differs significantly from the horizontal line, the data set is considered rhythmic. Regarding the p-values, the CircWave analysis only display the exact p-value in the case of circadian rhythmicity. Otherwise, the graph exhibited is a strait line and no p-value is presented.

The statistical analiysis por and it is not clear. The graph indicate 4 points by hour. However, ANOVA, Kruskal Wallis or not parametric analysis are not present.

AR: ANOVA was used initially to determine whether there were significant differences between the different timepoints. However, this information is irrelevant when it comes to rhythmicity. Normally, a significant difference detected by ANOVA is indicative of the presence of circadian rhythmicity, but there are also situations in which there are no significant differences according to ANOVA and the data still shows circadian rhythmicity when analyzed using CircWave and vice versa.

-The graph does not have statistical errors or media error standard. Please, correct it.

AR: The graphs currently show the various points at each time point and their corresponding average according to the CircWave analysis. In our opinion, the graphs with error bars will make the analysis more confusing. Here's an example (Figure 1). If the reviewer considers it relevant, all the graphs with the error bars will be included in the manuscript.

Figure 1. A representative graph with the error bars included.

Line 257. Please, explain (details) about how The ΔCt was calculated using the housekeeping gene as the reference gene, and the ΔΔCt was calculated between the normalized ΔCt values from each time point and the average Ct of all the time points tested.

AR: To calculate ΔCt we used the average of the Cts of a sample from which we subtracted the average of the Cts of the housekeeping gene for the same sample. To calculate ΔΔCt we first subtract the average of the ΔCt from the average of the Cts of the housekeeping gene of all the samples. Then we subtract the previous result from the ΔCt of the desired sample. This information was included in the manuscript.

Line 257. Please, explain the reason by which we use 100nM dexamethasone for the celular synchronizedion.

AR: Glucocorticoids and especially dexamethasone have been described as having the ability to synchronize the molecular clock [5]. This information was included in the manuscript.

The role of rAbcg2 in the transport of DNPZ across the BCSFB was analyzed, using  a similar assay with an inhibitor of rAbcg2 (Ko143 100nM; Tebu-bio, Lisbon, Portugal). Please, indicate all graphs of HPLC for treatments.

AR: A representative graph with the inhibitor was added to the manuscript.

-Indicate details on the procedure for the validation of the described procedure by the principles of the Food 302 and Drug Administration, cyte 26 [26].

AR: For this validation, FDA criteria were followed. Nine calibrators (n=5) were established within the linearity range between 0.04-40 μg/mL, and additionally, four quality controls (0.04, 1.25, 10, and 40 μg/mL) (n=3) were included. The criteria used to assess the fitness of this linear model included a weighted determination coefficient (R^2) higher than 0.99, and the accuracy of the calibrators within ± 15% from the nominal value (except at the LLOQ, where ± 20% was accepted) was adopted as acceptance criteria. The method's LLOQ was defined as the lowest concentration that could be precisely and accurately measured, with a coefficient of variation (CV) equal to or lower than 20% and a relative error (RE) within ± 20% of the nominal concentration. To evaluate sensitivity, the limit of detection (LOD) as a signal-to-noise ratio >3 was calculated, with 10 replicates performed at a concentration of 0.04 μg/mL.

Precision and accuracy were evaluated during the 5-day protocol adopting the same concentrations used for the quality controls. Coefficients of variation (CV) equal to or lower than 15% were accepted for precision at all studied concentration levels, while for accuracy, a mean relative error (RE) of ± 15% (from the nominal concentration) was accepted for all concentrations, except the LLOQ (± 20%).

The CVs obtained in the study of inter-day precision and accuracy (RE) were typically lower than 8%, with an accuracy ranging from 0.3 to 3.4% (Table x). As for intra-day precision and accuracy, it was evaluated on the same day by the analysis of five replicates at 0.04 (LLOQ); 1.25; 10 and 40 μg/mL. The obtained CVs were once again within the accepted criteria, with the CVs lower than ± 14%, and accuracy ranging from 0.5 to 14% (Table x).

Table x: Inter-day (n = 5), intra-day (n = 5) precision and accuracy

Concentration (μg/mL)

Inter-day precision

Intra-day precision

Measured*

CV (%)

RE

Measured*

CV (%)

RE

0.04

0.04 ± 0.003

7.24

3.94

0.04 ± 0.005

13.78

-2.00

1.25

1.22 ± 0.050

4.07

-2.40

1.24 ± 0.056

4.53

-0.48

10

9.97 ± 0.652

6.54

-0.32

11.35 ± 0.616

5.43

13.54

40

39.80 ± 2.195

5.52

-0.51

40.96 ± 2.256

5.51

2.40

*Mean values ± standard deviation

This information was added to the manuscript

Explain how outliders were calculated in your study

AR: Outliers were calculated taking into account these FDA criteria at the level of each parameter, considering the permitted CV and RE. This information was added to the manuscript.

Minnor comments

Line 299. The cromatogram for DNPZ . Chromatogram obtained for Donepezil (10µg/mL). Chromatogram for Donepezil  (10µg/mL) must also indicate curves with the inhibitor (show representative figures)

AR: Representative curves were included in the manuscript.

[1]         Schwerk C, Tenenbaum T, Kim KS, Schroten H (2015) The choroid plexus-a multi-role player during infectious diseases of the CNS. Front Cell Neurosci 9, 80.

[2]         Ballesta A, Dulong S, Abbara C, Cohen B, Okyar A, Clairambault J, Levi F (2011) A combined experimental and mathematical approach for molecular-based optimization of irinotecan circadian delivery. PLoS Comput Biol 7, e1002143.

[3]         Balakrishnan A, Stearns AT, Rounds J, Irani J, Giuffrida M, Rhoads DB, Ashley SW, Tavakkolizadeh A (2008) Diurnal rhythmicity in glucose uptake is mediated by temporal periodicity in the expression of the sodium-glucose cotransporter (SGLT1). Surgery 143, 813-818.

[4]         Zhang YK, Yeager RL, Klaassen CD (2009) Circadian expression profiles of drug-processing genes and transcription factors in mouse liver. Drug Metab Dispos 37, 106-115.

[5]         Balsalobre A, Brown SA, Marcacci L, Tronche F, Kellendonk C, Reichardt HM, Schutz G, Schibler U (2000) Resetting of circadian time in peripheral tissues by glucocorticoid signaling. Science 289, 2344-2347.

Round 2

Reviewer 2 Report

Comments and Suggestions for Authors

Dear Authors

My requirement has not been solved in this version. This in vitro model does not resemble characterisitic of circadian clock in culture, which is affected by more factors than glucocorticoid (DEX 100 nM) alone. Some figures are difficult to follow in this version again.

My Decision is Reject the manuscript

Comments on the Quality of English Language

The english style can be improved.